# Period Poverty in Brazil: A Public Health Emergency

**DOI:** 10.3390/healthcare13161944

**Published:** 2025-08-08

**Authors:** Maurício Fonseca Ribeiro Carvalho de Moraes, Rui Nunes, Ivone Duarte

**Affiliations:** Center of Bioethics, Faculty of Medicine, University of Porto, 4200-450 Porto, Portugal; ruinunes@med.up.pt (R.N.); iduarte@med.up.pt (I.D.)

**Keywords:** gender equality, menstrual health and hygiene (MHH), menstrual insecurity, period, SDGs

## Abstract

Period poverty is a broad and complex issue that intersects with various areas, including health, education, infrastructure, and human rights, among others, affecting countless women and girls around the world. Despite remarkable technological, social, and economic advances this century, menstruation remains a taboo subject, which leads to widespread misinformation and stigma. Prejudice and a lack of access to knowledge and essential sanitation resources, such as clean water, hygiene products, and safe private spaces, heighten the vulnerability of those affected. Integrated and multisectoral approaches that involve legislature, health, education, and sanitation are necessary to face this public health issue effectively. These efforts involve developing and implementing comprehensive plans that unite government, society, and the private sector. Some examples of these actions include making information about menstruation and menstrual health available in schools, cutting taxes on feminine hygiene products, improving basic sanitation, building decent public restrooms, and providing free sanitary pads in schools and workplaces. These initiatives have the potential to promote menstrual health and dignity, ensuring that people who menstruate can manage their periods in healthy, safe, and supportive environments. This review aims to shed light on menstrual poverty in Brazil as a global issue and a human rights violation, especially when it comes to the rights to health, education, and dignity. It stresses that efforts to end this social stigma and align with the 2030 Agenda, which seeks to eliminate poverty and inequality worldwide, and provides a plan of action to tackle this stigma.

## 1. Introduction

Menstrual health and hygiene are crucial for reproductive health and are acknowledged as basic human rights [1,2]. Menstrual health refers to overall physical, mental, and social well-being related to the menstrual cycle, highlighting the complex nature of menstruation and the various ways it can impact the lives of those who menstruate [3,4]. According to the World Bank [5], at least 500 million worldwide, including young girls, women, and nonbinary individuals from low-income, rural, or marginalized communities, lack adequate facilities for menstrual hygiene management (MHM). Inadequate water, sanitation, and hygiene (WASH) facilities, particularly in public places such as schools, workplaces, or health centers, can pose a major obstacle for women and girls in managing their periods safely and with dignity [6,7].

Limited access to menstrual products, clean water, proper healthcare, safe sanitation, and education about menstruation, combined with the ongoing stigma around periods, has led to what is now known as period poverty [8]. For example, adolescent girls in India began menstruating without understanding its cause, faced various restrictions, and often relied on cloth, especially in rural areas [9,10]. At the same time, commercial pads were more common in urban or school settings. This issue not only undermines fundamental human rights but also impacts the daily lives and futures of those affected.

Period poverty highlights the female body as a site of systemic vulnerability. Despite menstruation being a biological process shared by half the population, it continues to be seen as taboo in many societies, wrapped in shame, silence, and misinformation that urgently needs to be dismantled [11,12]. These sociocultural barriers contribute to poor menstrual literacy and prevent the normalization of menstruation in education, health, and policy discussions.

The implications of period poverty extend far beyond the individual, penetrating deeply into the fabric of society. In regions with high economic vulnerability, countless girls and women are forced to resort to dangerous alternatives, such as old rags, paper, or even leaves, to manage their menstrual cycles. This dire situation often leads to severe urogenital infections, overwhelming emotional distress, frequent absenteeism from school or work, and loss of income [13,14,15,16,17,18].

Educational disruption is particularly alarming: many girls experience menarche without essential knowledge or support, resulting in feelings of fear, shame, and the heartbreaking potential of dropping out of school [15]. This lack of education not only undermines their personal development but also cripples their ability to contribute economically. When menstruating women face significant discomfort and embarrassment, which often leads to job loss, the repercussions ripple through families and communities. Women, frequently seen as the backbone of the household, suffer financially due to lost workdays, further exacerbating the cycle of poverty and economic instability [19].

Furthermore, the emotional burden can be crushing. The stigma surrounding menstruation leads to feelings of stress, social isolation, depression, and anxiety, leaving these women in silence and suffering [7,18]. Despite being a long-standing problem, period poverty has only recently gained wider attention as its public health, social, and economic impacts have become clearer. There is a growing understanding that tackling this issue requires more than just distributing products; it demands thorough education, proper infrastructure, cultural change, and policy reform. Attaining menstrual equity is crucial not only for physical well-being but also for social justice and gender equality [16,17].

## 2. Period Poverty Worldwide

Period poverty is a global, intersectional issue that affects approximately 500 million people worldwide [20,21]. Far from being merely a hygiene concern, it intersects with broader public health inequities, gender-based discrimination, social exclusion, and economic vulnerability [22,23]. Its effects are felt across countries of all income levels, impacting girls, women, and non-binary individuals in both urban and rural contexts.

In many low- and middle-income countries, the consequences of period poverty are dire. For example, in Kenya, 7% of women and girls use unsafe menstrual materials like rags, feathers, and newspapers because they cannot access disposable pads, which are only available to 46% of the population [20]. In Kibera, Nairobi, 65% of women have engaged in transactional sex to afford sanitary products, exposing them to exploitation, violence, and health risks [24]. Similarly, in Bangladesh, the HER Project reported that 73% of women in the textile industry missed up to six workdays per month due to menstrual stigma and lack of products. Post-intervention, absenteeism dropped to just 3%, showing the impact of access and education [21].

In West and Central Africa, many women lack safe and hygienic workplace sanitation facilities, affecting productivity and well-being. In Ghana, 11.5 million women face inadequate menstrual hygiene infrastructure, contributing to continued exclusion and health hazards [25].

The crisis is not confined to low-income countries. In the United States, menstrual products are not covered by major public assistance programs like WIC and SNAP, despite being essential for dignity and health. Many low-income women miss work due to lack of access, while also paying a “pink tax”: in New York, for example, women’s hygiene products cost 13% more than men’s [26]. In Portugal, women pay more than men for nearly 42% of products, and in Brazil, that figure reaches 49.35%, with men paying more in only 3.9% of cases [23].

These financial and social burdens are compounded by widespread misinformation and taboos around menstruation. Women and girls report feeling shame, stress, and isolation, contributing to mental health challenges such as depression and anxiety [27]. These experiences are especially acute in environments where basic needs like potable water and safe bathrooms are also lacking.

The historical underrepresentation of women in biomedical research further exacerbates this issue. Up to 64% of sex-disaggregated health interventions show lower efficacy or accessibility for women, directly impacting their well-being and economic participation. As a result, women spend 25% more of their lives in poor health than men, contributing to an estimated USD 1 trillion in global productivity losses each year [23].

According to UNFPA and WHO, MHM must be recognized as a fundamental public health need. The lack of access to adequate menstrual products, private and hygienic facilities, accurate information, and supportive healthcare not only compromises individual dignity but also constitutes a violation of basic human rights [2,25]. Sommer et al. [28] highlight that millions of school-aged girls in low- and middle-income countries face barriers to managing menstruation safely and with dignity. These barriers include insufficient guidance before menarche, fear and stigma, inadequate sanitation in schools, lack of soap, privacy, safe disposal options, and even engaging in transactional sex for menstrual materials. The consequences profoundly interfere with girls’ education, self-esteem, health, and overall well-being, making MHM a matter of equity, safety, and social justice. A coordinated, multisectoral response across education, health, sanitation, and gender equity is essential for girls to thrive in school and life.

To effectively tackle period poverty, we must dismantle social taboos, eliminate financial barriers (such as taxes on menstrual products), and invest in gender-sensitive research and policies. Menstrual equity is crucial not only for achieving gender equality and educational attainment but also for fostering economic development. The scientific literature and human rights frameworks converge on this point: any strategy for addressing the social determinants of health must regard menstruation as a vital component.

## 3. Period Poverty in Brazil

Period poverty in Brazil is a multidimensional issue rooted in structural inequalities, characterized by socioeconomic, racial, and territorial disparities. This condition disproportionately impacts adolescents and women from marginalized communities, including rural areas, urban peripheries, and vulnerable social contexts [20,29].

The severity of menstrual precariousness is intensified by a lack of comprehensive data and broader social neglect. Approximately 13.6 million Brazilians (about 6.5% of the population) live in extreme poverty, surviving on less than USD 1.90 per day (around BRL 151 per month, based on the 2019 exchange rate), while roughly 51.5 million people (one in four Brazilians) fall below the poverty line, living on less than BRL 436 monthly [30,31].

Data from the Instituto Trata Brasil, in collaboration with BRK Ambiental, highlight stark inequities in water access, indicating that one in four women in Brazil lacks regular access to treated water. Alarmingly, between 2016 and 2019, the proportion of women without home bathrooms increased by 56.3%, affecting 2.5 million women, severely impacting their physical, mental, and social well-being [32].

Further disparities are evident when examining water sources for girls. Approximately 2.8 million girls (18%) reside in homes that rely on wells, stored rainwater, or other alternative water sources, and 2.3 million girls receive water irregularly, even when connected to the general supply network [33,34].

Sanitation infrastructure exacerbates these conditions, with a 15.5% annual increase in the number of women residing in homes lacking sewage collection, rising from 26.9 million in 2016 to 41.4 million in 2019 [32]. Additionally, 6.5 million girls live in households without proper sewage disposal, relying on ditches, unconnected cesspits, rivers, lakes, or the sea. Racial disparities are particularly pronounced: nearly 37% of Black girls compared to approximately 24% of White girls live in homes without sewage connections [33,34].

Regarding essential WASH conditions for menstrual hygiene, the 2017–2018 Household Budget Survey (POF) identified significant deficits. Over 900,000 girls (5.84%) live in homes without indoor piped water, while an even more vulnerable group, over 570,000 girls (3.7%), lack access to piped water entirely, including external property locations [33,34].

The school environment further compounds menstrual poverty. The PeNSE survey [35] revealed only São Paulo and the Federal District offer universal bathroom access in schools. In contrast, states like Maranhão and Roraima report over 10% of schools lacking restrooms. Data from the 2018 School Census indicate nearly half of Brazilian schools lack connection to a sewage system, and more than a quarter lack public water supplies. In northern Brazil, less than 10% of schools have adequate sanitary sewage systems [36]. Consequently, around 4 million girls face hygiene deprivation in schools, with nearly 200,000 lacking basic menstrual management resources. Many schools lack fundamental hygiene materials: 8% have no toilet paper, 4% no sinks, and 37% no soap. Black girls disproportionately experience these inadequacies, comprising over 60% of those in the most precarious conditions [37].

Economic disparities further underscore menstrual inequity. According to POF data, 8.7 million households with girls report sanitary pad purchases, but the financial burden varies significantly by socioeconomic status. Households in the poorest 20% spend an average of BRL 3.75 per unit of consumption monthly on sanitary products, compared to BRL 28.44 among the richest 20%. Direct expenditures by girls aged 10–19 average BRL 3.82 for the poorest quintile versus BRL 14.17 for the richest quintile. Overall, average spending per consumption unit is BRL 12.97, which drops to BRL 5.92 per woman when considering all reproductive-aged women in a household. This figure is even lower for Black girls, averaging BRL 5.45, which is 18% less than their White counterparts [33,34].

Furthermore, the tax burden exacerbates economic barriers, as menstrual products remain classified as non-essential items. Motta and Brito [38] demonstrated that this taxation discriminates against people who menstruate, hindering access and affecting gender equity. Miyake [39] criticized the selectivity and pointed out its consequences for school dropouts and access to menstrual health. For the poorest 5% of Brazilian women, lifetime menstrual product costs equate to approximately four years of income, forcing reliance on inadequate alternatives such as cloth, toilet paper, or improvised materials [40].

Cultural stigmas around menstruation further compound these difficulties. More than half of Brazilian adolescents receive no education on menstruation before their first period, fostering shame, misinformation, and social exclusion [34,41,42]. Menegotto and Ribeiro [43], through ethnographic research in Porto Alegre, demonstrated how intersections of race, gender, and socioeconomic class deepen menstrual poverty. Girls frequently experience racist, sexist, or ageist prejudices in healthcare settings, intensifying their vulnerability.

Moraes et al. [44] highlight institutional invisibility in relation to menstrual health, pointing out significant gaps in intersectoral policies, including health, education, and sanitation. Thus, such initiatives are largely limited to distributing sanitary pads, ignoring the right to information and the development of inclusive educational practices, which reinforces gender inequalities.

Research by SEMPRE LIVRE^®®^, Instituto Kyra, and Mosaiclab [45], with 814 low-income Brazilian women aged 14–45, found that 28% suffer from menstrual poverty. Many use unsafe materials like newspapers, socks, plastic bags, or coffee filters. In the past year, 28% suffered urinary tract infections, 24% had candidiasis, and many reported psychological impacts from inadequate menstrual management. Notably, 22% felt vulnerable during menstruation, and 9% lacked home bathroom access. The average cost of menstrual products is BRL 21, creating a significant financial burden; 21% reported difficulty purchasing them monthly [45].

In São Paulo, Soeiro et al. [46] reported that 42% of menstruating users of public health services do not have regular access to menstrual products, with 71.5% forced to reuse items, increasing the risk to reproductive health. Among vulnerable migrant populations—especially adolescent and adult Venezuelan women on the border—period poverty is more pronounced: in Boa Vista (Roraima), 46.4% of young migrants did not receive hygiene kits, 61% did not have access to basics such as water and sinks to wash their hands, and 75.9% felt unsafe in bathrooms, highlighting significant health risks [47,48].

Finally, menstrual poverty significantly disrupts educational outcomes, with 62% of Brazilian girls reporting school absenteeism due to menstrual-related issues, leading to increased dropout rates and diminished academic performance [45]. Menstrual poverty in Brazil is a crucial intersectional issue that spans public health, social justice, and human rights. It is a key indicator of social vulnerability, highlighting the unequal dynamics of social relationships [49]. To address this, we need robust, multisectoral policies across education, sanitation, healthcare, and fiscal reform to ensure menstrual dignity for everyone. Period dignity is recognized globally as a fundamental human right, emphasizing universal access to menstrual products and facilities to manage menstruation with dignity, regardless of socioeconomic status [50,51].

## 4. Strategic Alignment with Key SDGs: Deepening the Impact

Menstrual health is a fundamental human rights issue, critically under-addressed globally. According to the WSSCC, neglecting menstrual hygiene needs constitutes a violation of essential rights, including dignity, equality, bodily integrity, health, privacy, and freedom from inhumane or degrading treatment [21,52]. Ensuring menstruating individuals can live with dignity requires access to vital resources: reliable sanitation facilities, clean water, affordable menstrual products (disposable or reusable pads, menstrual cups), and psychosocial support to manage physical and emotional challenges, including cramps and mood swings [21,52].

International legal and policy frameworks underscore the urgency of addressing menstrual health. Instruments such as the Convention on the Rights of the Child (CRC) and the Convention on the Elimination of All Forms of Discrimination Against Women (CEDAW) emphasize informed decision-making and access to essential services and materials for menstrual health [53]. The WHO and UNICEF similarly recognize menstrual health and hygiene (MHH) as essential prerequisites and indicators for fulfilling core human rights, including education, employment, clean water, sanitation, health, and adequate housing [53].

Despite these international commitments, significant gaps persist. Many vulnerable communities continue to rely on unsafe materials, such as old rags or newspapers for menstrual management, posing severe health risks like urogenital infections, reproductive tract diseases, and cervical cancer [50,51,52,53,54]. These inadequate practices negatively impact physical health, education, economic participation, and emotional well-being.

Awareness gaps and insufficient structural support exacerbate the issue. Menstrual health remains stigmatized, leaving millions uninformed about managing menstruation safely, comfortably, and with dignity. Although human dignity is legally protected, menstruating individuals experiencing poverty, discrimination, and social silence often do not equally benefit from these protections. Cultural stigma, inadequate infrastructure, and limited education perpetuate persistent inequalities. Breaking this silence is crucial. Empowering religious, community, and institutional leaders to speak openly about menstruation can dismantle stigma and misinformation, fostering informed, autonomous decisions about health and well-being among menstruators [21,55].

Menstrual health aligns strategically with the 2030 Agenda for Sustainable Development Goals (SDGs) [56]. Addressing menstrual health intersects with multiple SDGs, providing a coherent framework for cross-sectoral actions to eradicate period poverty. This strategic alignment guides governments, institutions, and communities in formulating integrated policies to promote menstrual health through enhanced education, sanitation, gender equity, and economic inclusion, significantly improving the effectiveness of policymaking and advocacy.

SDG 1—No Poverty. Menstrual poverty arises from economic inequalities. Many menstruators cannot afford menstrual products, frequently taxed as non-essential items, placing substantial financial strain on low-income households. Providing free or affordable menstrual products can alleviate financial burdens, enabling economic stability and empowerment, particularly for girls, single mothers, and informal workers.

SDG 3—Good Health and Well-being. Poor menstrual hygiene contributes to health issues such as reproductive infections, cervical cancer risks, and psychological distress. Addressing menstrual poverty through sanitary product availability, clean water access, and health education significantly improves physical and mental health outcomes. Menstrual health literacy supports timely care-seeking behaviors and informed decision-making, reducing health disparities among marginalized populations.

SDG 4—Quality Education. Limited access to menstrual products and poor sanitation increase school absenteeism among girls. Fear and embarrassment often lead to lower academic performance and dropouts. Implementing gender-sensitive sanitation, providing menstrual products at schools, and offering comprehensive menstrual education can improve attendance, retention, and gender equality, especially in rural and disadvantaged areas regions.

SDG 5—Gender Equality. Period poverty arises from gender discrimination and stigma surrounding menstruation, which creates barriers in education, employment, and public life. Ensuring menstrual dignity requires addressing structural inequalities by eliminating stigma, empowering menstruators, and supporting inclusive policies that consider transgender and non-binary individuals. Tackling menstrual poverty directly aids in dismantling systemic gender inequalities, affirming bodily autonomy, and upholding reproductive rights.

SDG 6—Clean Water and Sanitation. Effective menstrual health management relies on strong water and sanitation infrastructure. Without clean water, private sanitation facilities, and proper waste disposal, menstrual hygiene becomes challenging. Poor WASH conditions in schools, public facilities, slums, and rural areas disrupt menstrual management, leading to absenteeism, health risks, and gender inequality. Prioritizing menstrual-friendly sanitation infrastructure improves dignity, safety, and privacy, enhancing health, education, and gender equality outcomes.

Addressing menstrual health comprehensively through strategic alignment with the SDGs presents a transformative opportunity. Integrating policy, infrastructure, education, and advocacy efforts can reduce menstrual poverty, improve quality of life, and foster inclusive development, highlighting menstrual dignity as essential to achieving sustainable development goals globally.

## 5. Efforts to Reduce Period Poverty in Brazil

Effectively addressing period poverty in Brazil requires integrated, multisectoral, and inclusive strategies rooted in human rights, equity, and period dignity. The issue extends far beyond the lack of sanitary products; period poverty reflects structural inequalities involving gender, race, class, and territorial exclusion, and must be treated as a central public policy concern [45].

A robust response must involve coordinated action across health systems, educational institutions, legislative frameworks, community movements, and private initiatives. Public investment in WASH infrastructure—especially in public schools and underserved areas—is essential to ensure that menstruation does not impede educational attendance, health, or civic participation. According to Andargie and Tinuola [57], school-based MHM and WASH interventions in Ethiopia significantly empower menstruating girls, enhance their physical, emotional, and social health, increase school attendance during menstruation, and reduce menstrual-related absenteeism. These efforts also prevent reproductive and urogenital infections, promoting educational equity and social inclusion. Initiatives like this exemplify a model with great potential for application in Brazilian schools.

Comprehensive menstrual education is essential. Integrating menstrual literacy into the national curriculum, as suggested by the ‘Cycle of Dignity’ [41], breaks the cycle of silence and misinformation that reinforces stigma and exclusion across school subjects. Without teacher training and culturally relevant materials, distributing sanitary pads lacks depth. Workshops with grassroots and LGBTQIA+ focus show that sustained campaigns can normalize menstruation in both formal and informal spaces, promoting intersectional inclusion.

Brazil has made notable legislative advances in recent years. Law No. 14.214/2021 [58], established from PL 4968/2019 [59], created the National Menstrual Health Protection and Promotion Program. It mandates the distribution of free menstrual products to public school students, incarcerated persons, and people living in homelessness. Additionally, laws such as Norm 6.779/2021 [60] and Rio de Janeiro’s Law 8.924/2020 [61] have included menstrual items in basic goods baskets and institutionalized menstrual education in public schools. However, implementation remains fragmented and uneven across states and municipalities.

Recent government data show that nearly 2 million people accessed free menstrual products through federal programs in 2023 [62]. Despite these advancements, logistical challenges, regional disparities, and the use of digital platforms like Meu SUS Digital remain obstacles, particularly in remote areas and among populations with limited literacy. Additionally, the exclusion of trans and non-binary individuals continues: many menstruation apps do not account for non-binary identities or inclusive algorithms; SUS health information systems still do not track gender identities beyond cisgender; and menstrual policies that equate menstruation with “women” end up erasing these identities [42,63].

Local initiatives like Menstruação Sem Tabu (São Paulo) and Dignidade Íntima (Ceará) show increasing local involvement. However, as Prado [64] points out, many of these programs follow a well-meaning welfare model. Still, they are focused on distributing products rather than providing systematic education, long-term institutional support, or adequate infrastructure, which does not amount to a structural solution. Period poverty is not merely a logistical issue; it highlights the historical marginalization of menstruation in public health agendas and requires a rights-based, justice-oriented approach [65].

Civil society has been instrumental in transforming public discourse and advocating for menstrual equity. Organizations such as Girl Up Brasil [66] and Project “Deixa Fluir” [67] have organized donation campaigns, youth mobilizations, and policy advocacy, ensuring that the lived experiences of marginalized groups—Afro-Brazilian girls, Indigenous communities, and LGBTQIA+ individuals—are centered in the policy debate. Their initiatives highlight that long-lasting change requires listening to those most affected.

Young leaders like Cecília Giacomin show innovative and sustainable approaches by promoting biodegradable pads and local production, which reduces environmental impact and creates income-generating opportunities for women. These community-based efforts combine menstrual justice with ecological and economic sustainability [68].

Academic institutions have likewise played a pivotal role. The Dignidade Menstrual project at UNIFESP and the work of Fegadolli and Uwai [69] demonstrate how rigorous research can inform evidence-based policies. Universities have been crucial in assessing gaps in access, evaluating policy implementation, and developing theoretical frameworks that connect menstruation to broader public health and social justice outcomes.

To build sustainable and inclusive solutions, the following actions are essential: eliminate the “pink tax” and classify menstrual products as essential health goods [70,71,72,73]; guarantee free menstrual products in schools, prisons, shelters, Indigenous territories, and underserved regions [74]; expand WASH facilities, especially in public schools and rural areas, ensuring privacy, clean water, soap, and sanitary pads [75]; incorporate menstrual health in school curricula and public campaigns, dismantling taboos and promoting body literacy [76,77]; ensure menstrual health policies are accessible to all, including trans men, non-binary people, and people with disabilities [73,74,75,76,77,78]; and establish national indicators and data collection systems to assess access, effectiveness, and equity [79,80,81].

Period poverty is not a peripheral issue, as it intersects with fundamental rights to education, health, and dignity. Addressing it is essential to fulfilling Brazil’s constitutional guarantees and advancing the Sustainable Development Goals (SDGs), particularly SDGs 1 (No Poverty), 3 (Good Health and Well-being), 4 (Quality Education), 5 (Gender Equality), and 6 (Clean Water and Sanitation). Ultimately, period dignity must be seen not as a privilege, but as a constitutional and human right. Only through sustained investment, inclusive policymaking, and active civic participation can Brazil move from fragmented responses to systemic solutions, ensuring a just and equitable future for all menstruating people.

## 6. Conclusions

Period poverty is a serious public health and human rights issue in Brazil and worldwide, reflecting deep inequalities affecting access to sanitary products, health, education, and gender equity. This paper shows that period poverty is rooted in structural injustices exacerbated by poverty, lack of infrastructure, stigma, and neglect. Its consequences impact physical and mental health, school attendance, labor participation, and social and economic growth. Notable advances, like the Menstrual Health Protection and Promotion Program, face persistent barriers such as the digital divide, implementation gaps, and exclusionary policies that hinder equal access to menstrual health resources. While this review highlights critical dimensions of period poverty in Brazil, further research is needed to evaluate the long-term efficacy of implemented policies, explore intersectional disparities (e.g., rural, Indigenous, and LGBTQIA+ communities), and develop standardized metrics for monitoring menstrual health interventions. Civil society, youth movements, and academia have transformed period poverty into a national policy issue, driving legal reforms, awareness, and innovative solutions. Brazil must recognize period dignity as a constitutional right and a policy priority. Ending period poverty requires investment in education, inclusive health policies, improved sanitation, and removal of fiscal and cultural barriers. Aligning these efforts with the Sustainable Development Goals—especially those related to poverty, health, education, gender equality, and sanitation—ensures menstruating individuals are not overlooked in national development. Addressing period poverty means not just distributing products but also changing systems that silence and exclude. It guarantees menstruation does not hinder anyone’s ability to learn, work, or fully participate in society. By advancing menstrual equity, Brazil can foster a more just and resilient future, where menstruation signifies progress and dignity for all.

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
