# Peer review of "Period Poverty in Brazil: A Public Health Emergency"

_healthcare, 2025, doi:10.3390/healthcare13161944_

Round 1
Reviewer 1 Report (Previous Reviewer 2)
Comments and Suggestions for Authors
The authors have adequately addressed the comments raised during the initial review. Revisions have significantly improved clarity, structure, and analytical depth. The focus on Brazil is now contextualized within a global framework, and the inclusion of actionable, multisectoral recommendations enhances its practical relevance. Key issues raised by reviewers have been resolved. A few minor edits are recommended, including standardizing the author information section, arranging keywords in alphabetical order, adding a sentence on study limitations and directions for future research, adding an abbreviation list .. Subject to these final refinements, the manuscript is suitable for publication in Healthcare.
Author Response
Editorial Office
Healthcare
MDPI
Subject: Submission of Revised Manuscript – “Period Poverty in Brazil: A Public Health Emergency”
Dear Editor,
I am pleased to submit our revised manuscript entitled “Period Poverty in Brazil: A Public Health Emergency”, for consideration in Healthcare. We addressed the comments and implemented the necessary revisions to enhance the manuscript’s quality and adherence to journal guidelines.
The key revisions include are: Standardization of author information; Reorganization of keywords; Incorporation of a new sentence in the conclusion; Addition of an abbreviation list. All the changes are highlighted in yellow.
Thank you for your time and consideration.
Best regards
Mauricio Fonseca Ribeiro Carvalho de Moraes
This manuscript is a resubmission of an earlier submission. The following is a list of the peer review reports and author responses from that submission.
Round 1
Reviewer 1 Report
Comments and Suggestions for Authors
The paper is an opinion-based literature review that addresses menstrual poverty. The study underscores the importance of comprehensive, multi-sectoral approaches to promote menstrual health, raise awareness, and empower individuals for overall well-being.
Here are a few comments:
- While the topic is crucial, there are instances that may mislead readers. For example, the content primarily focuses on Brazil, contrasting with the global concept suggested in the abstract.
- In the introduction, the authors state, "This discussion was based on a literature review of national and international articles on the web and in journals, sourced from studies and research conducted by competent bodies." While the paper is opinion-based, supported by a literature review, the introduction needs clarification.
- The headings and subheadings are confusing, lacking clear analysis. Consider revising the headers to better align with the content.
- There is a significant issue with the conclusions section, as much of the content seems more suited for the discussion. I recommend integrating this content into the discussion and consolidating the conclusions into a single paragraph.
A revision and resubmission of the paper would be beneficial.
Reviewer 2 Report
Comments and Suggestions for Authors
The manuscript addresses a critical issue with global relevance, providing an exploration of menstrual poverty, with references and a comprehensive understanding of its dimensions. The authors could consider strengthen it by offering clearer solutions, deeper analysis, and a more focused narrative rather than relying heavily on external reports. If these revisions are made, it could become a strong and influential paper on an important subject.
General comments:
· The manuscript shows several strengths, as it covers a wide range of issues related to menstrual poverty, effectively presents menstrual poverty as a public health, educational, and socio-economic issue. The inclusion of global statistics, comparisons between countries, and references to UNICEF, UNFPA… highlights the universality of the issues. The framing of menstrual poverty as a human rights issue is a powerful angle.
· The argument sometimes feels scattered due to the variety of subtopics (political, cultural, and economic) being addressed simultaneously without a strong unifying thread.
· While citing a broad range of sources, the manuscript could use more analysis of these references. It seems not offering enough discussion or original interpretation.
· The manuscript strongly presents the problems but offers relatively few solutions. While it calls for multisectoral approaches and legal changes, it does not provide enough practical recommendations.
· Some ideas are repeated throughout the manuscript, such as the link between menstrual poverty and human dignity.
· The manuscript references religious stigma (Leviticus) but does not fully explore how cultural and religious beliefs might hinder or help address menstrual poverty in different contexts.
· It seems lacking an analysis comparing the policies in different countries, or how cultural attitudes toward menstruation affect policy implementation…
· Ensure that all citations are formatted consistently. Italics are generally not used for citations themselves.
Specific comments:
The terms menstrual poverty and menstrual dignity are used frequently but inconsistently. Ensure that these terms are clearly defined early in the manuscript and used consistently throughout.
Lines 11-26: The abstract introduces menstrual poverty as a global issue but does not clarify whether the study focuses on a specific population, such as Brazil or globally.
The abstract mentions multi-sectoral approaches but does not provide details on what these solutions would look like or how they could be implemented. It may need to include at least one example.
Line 31-35: The reference to Leviticus is useful. However, this introduction feels somewhat abrupt. Consider expanding this point to better connect it to the broader issues of menstrual health education and societal stigma. A more nuanced transition into the scientific and modern-day context would help make this more cohesive.
Lines 33-34: The inclusion of Leviticus 15:24 feels somewhat out of place. While the historical context is valid, more explanation of why this reference is relevant today would strengthen its purpose.
Lines 45-47 and 57-61: The goal of study is repeated in multiple. Consider condensing these points into a single statement.
Line 51: "This discussion was based on a literature review …" does not cite specific papers or journals. There are substantial arguments without citations.
The introduction states the purpose of the study but lacks a clear thesis that ties the study's findings to specific outcomes or proposals.
Lines 170-179: The economic effects of menstrual poverty (absenteeism in Bangladesh) are mentioned, but this section could be enriched with further discussion of how addressing menstrual poverty could boost public health.
Lines 296-318: The reference to SDGs is important but somewhat generic. It would be to focus on a few key SDGs most relevant to menstrual poverty and explain in greater detail how addressing menstrual poverty contributes directly to these goals.
The authors could consider discussing some limitations of the study.
Line 327: United Nations Population Fund and the United Nations Children's Fund should be abbreviated. Please check consistent use of abbreviation throughout the manuscript.
Conclusion: The conclusion is broad and lacks specificity regarding what actions need to be taken by different stakeholders. Consider adding a clear set of actionable recommendations for governments, NGOs, and international organizations.
